# Efficacy of Personalized Foot Orthoses in Children with Flexible Flat Foot: Protocol for a Randomized Controlled Trial

**DOI:** 10.3390/jpm13081269

**Published:** 2023-08-17

**Authors:** Cristina Molina-García, Andrés Reinoso-Cobo, Jonathan Cortés-Martín, Eva Lopezosa-Reca, Ana Marchena-Rodriguez, George Banwell, Laura Ramos-Petersen

**Affiliations:** 1Health Sciences PhD Program, Universidad Católica de Murcia UCAM, Campus de los Jerónimos 135, 30107 Murcia, Spain; cmolina799@ucam.edu; 2Department of Nursing and Podiatry, Faculty of Health Sciences, University of Malaga, Arquitecto Francisco Peñalosa 3, Ampliación de Campus de Teatinos, 29071 Malaga, Spain; andreicob@uma.es (A.R.-C.); evalopezosa@uma.es (E.L.-R.); amarchena@uma.es (A.M.-R.); lauraramos.94@uma.es (L.R.-P.); 3Research Group CTS1068, Andalusia Research Plan, Junta de Andalucia, Nursing Department, Faculty of Health Sciences, University of Granada, 18071 Granada, Spain; jcortesmartin@ugr.es

**Keywords:** flexible flatfoot, pediatrics, children, foot orthosis, strengthening exercises

## Abstract

Pediatric flat foot (PFF) is a very frequent entity and a common concern for parents and health professionals. There is no established definition, diagnostic method, or clear treatment approach. There are multiple conservative and surgical treatments, the implantation of foot orthoses (FO) being the most used treatment. The evidence supporting FO is very thin. It is not clearly known what the effect of these is, nor when it is convenient to recommend them. The main objective of this protocol is to design a randomized controlled trial to determine if personalized FO, together with a specific exercise regimen, produce the same or better results regarding the signs and symptoms of PFF, compared to only specific exercises. In order to respond to the stated objectives, we have proposed a randomized controlled clinical trial, in which we intend to evaluate the efficacy of FO together with strengthening exercises, compared to a control group in which placebos will be implanted as FO treatment along with the same exercises as the experimental group. For this, four measurements will be taken throughout 18 months (pre-treatment, two during treatment and finally another post-treatment measurement). The combination of FO plus exercise is expected to improve the signs and symptoms (if present) of PFF compared to exercise alone and the placebo FO group. In addition, it is expected that in both conditions the biomechanics of the foot will improve compared to the initial measurements.

## 1. Introduction

Pediatric flat foot (PFF) is a very frequent syndrome in primary care consultations, and it is also a shared concern among parents and professionals. Currently, there is no clear definition for the diagnosis of PFF, no treatment protocol, nor solid scientific evidence on the wide range of treatments [1,2].

PFF is characterized by a talocalcaneal misalignment, which is reflected as a collapse of the medial longitudinal arch (MLA) in a standing position [3]. Consequently, there is excessive pronation, accompanied by a drooping of the navicular. Some feet are also accompanied by calcaneal valgus [4]. All these alterations in the morphology of the foot force the rest of the structures, such as soft tissues or joints, to compensate for the excessive forces that act on the MLA [5]. These compensations cause an inefficient gait and symptoms such as pain, fatigue, stumbling when walking, problems in the proximal joints and reduced quality of life [6,7].

Diagnosis is based on clinical findings, including a variety of clinical tests and radiographic signs, these being considered the “gold standard” [6,8,9].

Regarding treatment, there are both conservative and surgical options, surgical being the last treatment option [10]. The most common conservative treatment is the use of foot orthoses (FO), where previous studies have concluded that they improve the results of some clinical tests, radiographic angles and symptomatology [11,12,13,14,15,16,17]. The purpose of the FO is to modify the position of the axis of the subtalar joint, decrease the speed of pronation, support the MLA and distribute loads more effectively [18,19]. Previous systematic reviews indicate that FO are beneficial and create positive changes in the development of the child’s foot [8,20,21]; these changes being greater in earlier ages of the treatment [22]. The exact age to start treatment is not clear, although it is recommended to start at preschool age (under 7 years old) so that the possibility of correcting the PFF is greater [23,24]. It has also been seen that the combination of FO with exercises is much more beneficial [25]. However, other studies indicate that the modifications that occur are those resulting from the natural development of the foot [26,27]. The evidence regarding the use of FO still does not present a consensus [20,28,29].

Therefore, there is a discrepancy between treating and not treating PFF. There are authors who conclude that it is not necessary, since the natural evolution of the foot is that the MLA begins to form at 3–4 years of age and ends at 10 years of age [26,27]. A recent meta-analysis [30] concludes that, due to the normal development of the foot, treatments should be ruled out unless there are symptoms such as pain, limited function or reduced quality of life. However, other authors recommend early treatment, based on the fact that flat feet persist in 23% of adults and may be associated with Achilles tendinopathy, plantar fasciopathy, tibial posterior tendinopathy, hallux rigidus, chondromalacia patellae or patello-femoral pain syndrome [1,2,7,13,20]. Based on these latest data, it would be unethical to leave these types of feet untreated. Additionally, a recent systematic review [24] demonstrated that FO are beneficial, with evidence regarding efficacy in treating signs and symptoms.

Therefore, since PFF is a very common syndrome which, if left untreated, could cause problems in the long term and there is no consensus on the treatment protocol, it is necessary to investigate the effectiveness of FO in terms of improvement of the signs and symptoms, including the prevention of pathologies or injuries and the improvement of the quality of life. Sagat, P et al. have shown that children with flat feet presented poor performance in certain physical tasks, in contrast to a control group with neutral feet [31]. In addition, since it is a subject of great interest for researchers, health professionals and parents, it is necessary to carry out an investigation to assess the effectiveness of FO, with a standardized diagnostic protocol with validated tests in a larger sample size than previously published studies and with a longer-term follow-up. Furthermore, as the authors Zhang J. et al. pointed out, early identification of PFF is necessary; thus an intervention plays a crucial role in enhancing the outlook. Also, there is an absence of consistent quantitative standards for diagnosing flexible flatfoot [32].

Therefore, the objective of the current study is to design a protocol for a randomized controlled trial (RCT) to determine whether personalized FO together with a specific exercise regimen produce the same or better results regarding the signs and symptoms of PFF, compared to only specific exercises. In addition, as specific objectives, to detect whether the possible bias that has prevented previous studies from demonstrating the efficacy of FO is due to the fact that these FO were not personalized; to define the PFF and to evaluate if there is a correlation between the clinical methods and the diagnosis and severity of the PFF.

## 2. Materials and Methods

### 2.1. Study Design and Setting

The design is a randomized controlled two-arm trial.

Patients will be recruited from the university clinic from Universidad Católica San Antonio de Murcia, University of Malaga and schools nearby in Spain. They will be randomized to one of the two groups, each receiving a different intervention. The schedule to follow while carrying out this RCT can be found in Appendix A. The total period intended to be allocated to this study is from September 2023 to February 2025.

To randomize the sample, a Microsoft Excel spreadsheet will be used where a random assignment sequence will be generated. Each patient will be given a consecutive number in order of arrival and allocation concealed in envelopes.

### 2.2. Eligibility Criteria

Subjects aged 3 to 12 years diagnosed with PFF. For the diagnosis of the PFF, the following criteria must be met:Foot Posture Index (FPI) > 6 [33].Navicular drop > 10 mm [34].Relaxed calcaneal stance position (RCSP) 6° to 12° valgus [35].Pronation angle > 10° [36].Arch index > a 1.35 [37].Double/single heel rise test negative [38].Windlass test negative [39].

In addition, the signature of the parents or legal guardians with consent to participate in the study will be necessary (Appendix B).

Participants will be excluded if they have undergone any surgery in the lower limbs, have previously received treatment for PFF, present osteoarticular injury, foot fractures or in the lower limbs in the last 6 months, ankle sprain, asymmetry, systemic diseases with osteoarticular involvement that present symptoms in the lower limb with gait disturbance (for example, Perthes disease) or biomechanical alteration of the lower limb with repercussions on the foot and ankle. Children suffering from any type of neurological or systemic disability (cerebral palsy, Down Syndrome, clubfoot or equino-varus, …) will also be excluded.

### 2.3. Interventions

At the first visit, parents will be informed of their child’s current problem, treatments, and the existence of this study. In the case of accepting to participate in the study, the process will begin: all the variables will be carefully collected, the treatment will be established and an appointment will be made for follow-up at 6 months.

First, all the affiliation data will be collected and the PFF will be diagnosed.

#### 2.3.1. Group 1

For custom insoles, a mold will be taken using phenolic foam. To do this, the child will be asked to sit in a chair (to take the mold semi-load baring). The mold will be taken in a corrected position, that is, limiting the internal rotation of the tibia with one hand and the windlass mechanism will be performed to increase the MLA. The mold must remain neutral, so in the event that a varus or valgus print has emerged, this process will be repeated. Once we have the mold filled with plaster, two modifications will be made: a moderate medial heel skive (5 mm and an angulation of 15°) in the hindfoot and a slight inversion balance of 4° in the forefoot. Once the mold is prepared, the FO will be thermoformed; for this, we will use a 3 mm polypropylene, 25 Shore-A EVA as lining and 65 Shore-A EVA to stabilize the hindfoot.

In addition, the exercises to be performed will be explained to them and they will be given all the recommendations regarding the performance of exercises, the use of FOs and shoe therapy. These explanations will be provided in an additional report (Appendix C) that will be given to all subjects together with a calendar so that they write down all the days they perform the exercises with an X.

The exercises will first be explained by the podiatrist to the child and the parent/legal guardian. In addition, a standard video will be created so that the child can see it, in which the exercises will be explained through drawings in order to capture the attention of the child. The parent/legal guardian needs to confirm that the child is doing it correctly.

Once the FO have been provided, after assessing that it adapts well to the foot and does not cause discomfort, the patient will be requested to come in at 6 months. In the event that there is any discomfort or irritation to the child’s skin, an appointment will be made to readjust or modify the FO (in this visit the variables will not be evaluated, only the FO will be fixed; if there are no problems, this visit will not be carried out).

#### 2.3.2. Group 2

The participants will have the same intervention described for group 1, with the difference that they will use a placebo FO. This will be constructed using flat 1 mm 65 shore Ethylene-vinyl acetate (EVA), which will be cut to the foot size of the child and covered with the same top cover as the FO Group 1 to visually prevent them from being distinguished.

The instruments necessary for the development of this study and the budget necessary to carry out this clinical trial are included in Appendix D.

### 2.4. Outcomes Measures

The document presented in Appendix C will be used for data collection. It contains all the data to be collected in the anamnesis and all the variables to be studied, including all the clinical tests that would be carried out.

#### 2.4.1. Qualitative or Categorical Variables:

Gender: masculine or feminine.Pain: symptomatic or asymptomatic.Level of physical activity: high-low-nil.Double/simple heel rise test: Standing on toes with two legs/one leg for 25 repetitions. It will be considered positive if the participant is incapable due to fatigue or if when raising the calcaneus does not present a varus position [38].Supination resistance test: high-moderate-low. The patient is instructed to stand relaxed without any attempt to move the foot or lift the arch. The examiner’s fingertips are then placed plantar to the medial half of the navicular, and the examiner exerts a significant lifting force on the navicular. A normal foot will demonstrate subtalar joint supination with minimal lifting force. A pes valgus deformity will need extreme amounts of lifting force in order to produce little, if any, subtalar joint supination motion [40].Subtalar joint axis: Lateralized-neutral-medialized. The center of the neck of the talus should be located and marked to see the lateralized or medialized point, or if, on the contrary, it stops at the 2nd finger, which would indicate that it is neutral [36].Shoe wear at heel level: medial-center-lateral.Maximum pronation test: Positive or negative. The patient is asked to try pronate as much as possible; it is considered positive when performing the maneuver, the calcaneus cannot pronate more than 2° [41].Forefoot: adduction-neutral-abduction position [36].Foot posture index (FPI): Normal = 0 to +5; pronated = +6 to +9; highly Pronated = +10 to +12; supinated = −1 to −4 and highly supinated = −5 to −12. The six clinical criteria assessed: 1. palpation of the talus head; 2. lateral supra and inframalleolar curvature; 3. position of the calcaneus in the frontal plane; 4. prominence of the talonavicular region; 5. congruence of the internal longitudinal arch and 6. abduction/adduction of the forefoot with respect to the rearfoot. As we observe them, the following score is given: neutral = 0; clear signs of supination = −2; clear signs of pronation = +2 [33].Test of windlass: Positive or negative. It will be considered positive if, when performing dorsiflexion of the hallux, there is not supination of the foot, plantarflexion of the 1st ray, increase in the MLA and internal rotation of the tibia [39].Beighton scale: Hypermobility or normal. Subjects are rated on a 9-point scale, considering 1 point for each hypermobile site. These 9 points are: 1-hyperextension of the elbows (more than 10°), 2-passively touch the forearm with the thumb, having the wrist in flexion, 3-passive extension of the index finger to more than 90°, with the palm of the hand resting on the bed, 4-hyperextension of the knees (10° or more), patient in supine position and 5-flexion of the trunk forward touching the ground with the palms of the hands by bending without bending your knees. To be considered as hypermobile, it is required to have 4 points or more of the total of 9 [42].Podoscope: pronated-supinated-neutral [36].Pressure platform: maximum pressure zone, location of the center of gravity, gait progression line [43].

#### 2.4.2. Quantitative or Numerical Variables

Age: in months.Weight: in Kg.Height: in meters.Body Mass Index (BMI): Will be calculated with the formula weight (Kg) divided by height squared (meters^2^). The classification of each child in low weight, normal weight, overweight or obesity will depend on the child’s sex, height, weight and age [44].Pain: visual analog scale (from 1 to 10, 1 being minimum pain and 10 maximum pain).FPI: The six clinical criteria used in PFI are: 1. palpation of the talus head; 2. curvature supra and lateral inframaleolar region; 3. position of the calcaneus in the frontal plane; 4. prominence of the talonavicular region; 5. congruence of the internal longitudinal arch and 6. abduction/adduction of the forefoot with respect to the rearfoot. (Score: neutral = 0; clear signs of supination = −2; clear signs of pronation = +2) [33].RCSP: degrees of calcaneal eversion. The valgus degrees of the calcaneus are measured in bipedal support [35].Navicular drop: in millimeters. It measures the difference between the navicular position when the patient’s foot is in a neutral position and when the patient’s foot is in its normal position. It measures how many millimeters the medial tuberosity of the scaphoid has descended [34].Pronation angle: in degrees. To calculate the bisection of the distal third of the tibia with respect to the bisection of the calcaneus [36].Chippaux-Smirak index: in cm. On the footprint of the subject taken from a pedigraphy, the narrowest distance from the medial part of the foot (B) with the widest distance from the forefoot (A) must be measured. It is divided B/A [45].Pressure platform: percentage of load/weight on each foot and distribution of the same (anterior load, posterior load, load of the left and right foot) [43].Arch index: numerical scale. The patient’s footprint is taken with a pedigraphy, the toe area is excluded and a longitudinal line is drawn that goes from the center of the heel to the 2nd toe. A line is then drawn perpendicular to the 1st. Two lines are drawn perpendicular to this axis to see the anterior extent of the forefoot area. The axis of the foot is divided into 3 equal parts and here 3 zones are defined: A: forefoot, B: midfoot and C: rearfoot. The arch index is calculated: B/(A + B + C) [37].Foot size: in cm.Silfverskiold test: in degrees. The degrees of dorsiflexion of the ankle (starting from a position of 90°) with extended knee and bent knee [46] will be measured.Navicular height: in millimeters. Measure the height of the scaphoid to the ground with the subject sitting [47].

### 2.5. Blinding and Monitoring

Participants and their parents/guardians will be blinded as to which group they are allocated to and will not see the other group.

An initial assessment will be made, these same measurements will be repeated 3 times throughout the duration of the study. In total we will obtain 4 measurements for statistical analysis.

The second measurement will be a month of treatment; in addition, in this visit we will assess the state of the FO and verify that the exercises are going well. We will ask about the FO, if any pain has appeared that was not there before, any blisters or reddened areas. In the event that the adaptation to the FO has not been good, we will make the necessary adjustments to the FO, such as lowering the MLA. If any subject needs their FO to be modified, they will be called by telephone after 2 weeks to see the evolution; in the event that it has not improved, it will be cited to re-evaluate the FO.

To corroborate that the execution of the exercises is good, we will ask the participant to repeat them, and in case there is an exercise that is not being performed correctly, we will explain it again.

The third assessment will be in the middle of the treatment/study period, that is, at 6 months. At this stage, all the initial measurements/assessments will be conducted again, as well as the last one at 12 months.

The estimated time for the measurement assessments is one hour for the 1st visit and 30 min for the rest of the visits.

### 2.6. Sample Size

The calculation of the sample size has been carried out with the data analysis program EPIDAT https://www.sergas.es/Saude-publica/EPIDAT?language=es (accessed on 4 August 2023). For the calculation, a clinical variation of 2 and a standard deviation of 1 have been considered. A statistical power of 80% and a significance level of *p* < 0.05; 95% confidence level. The minimum size would be a sample of 128 subjects (64 in each group randomly distributed). Considering that the loss rate could be 30%, the final size should be 84 subjects in each group. The total sample size should be 168 subjects.

### 2.7. Statistical Analysis

The description of data will be calculated using the percentages and frequencies of the qualitative variables and for the quantitative variables, the standard deviation and the mean. In addition, in case of presenting high deviations, the measures of central tendency would be calculated, as is the case of the mean, median or mode. All this will be carried out in frequency distribution tables of different categories, using SPSS (IBM SPSS Statistics: V.28, USA).

It is intended to compare the dependent variables with the independent ones. The Kolgorov–Smirnof test (the adjustment to the normal of the distribution) will be used to check the normality of the quantitative variables; in the event that they follow a normal distribution, the following techniques would be used: linear regression or Pearson correlation for the comparison of quantitative variables, chi-square (X^2^) to compare qualitative variables and Student’s *t* or ANOVA to compare qualitative with quantitative variables. In the event that they do not conform to the normal, it will be calculated according to the case; the Wilcoxon test, the Kruskal–Wallis test, and the Mann–Whitney test.

To determine that the supposed differences between control group and experimental group are not due to a random error, but to a real difference, in the bivariate analysis a hypothesis test will be carried out. The significance level of p shall be 0.05.

### 2.8. Ethics and Dissemination

The study has been awarded ethical approval from the committee of the Universidad Católica San Antonia de Murcia (CE032213).

## 3. Expected Results

Personalized FO along with a specific exercise regimen are expected to produce better results for signs and symptoms of PFF compared to specific exercises alone. In addition, it will be detected if the possible bias that has made the previous studies not demonstrate the efficacy of FO is that these FO were not personalized.

After the completion of this study, in which a large number of tests will be analyzed, the research will provide a definition of the PFF according to the common findings presented by the sample, a definition that nowadays is non-existent. Finally, we can evaluate if there is a correlation between clinical methods and the diagnosis and severity of PFF, in order to make an accurate diagnosis.

It is anticipated that the study will provide valuable evidence for improvement of the treatment of PFF, as well as for the diagnosis and management of this entity.

The main future research which is required after this protocol study is to carry out the detailed RCT described in the present manuscript. Also, qualitative research to understand the experience of patients with PFF wearing an FO is required.

### 3.1. Limitations

The main limitation that can be found in this RCT is the involvement and adherence to the study by parents or legal guardians and children. Another limitation that should be mentioned is not having a control group that does not undergo any treatment, as PFF can lead to problems in the biomechanics of gait and in the development of future pathologies. Also, we cannot claim that all patients will be using their orthoses the whole period of our study. This is because the study will be undertaken in Spain, where there are very high temperatures in summer. This may make it difficult for the patients to wear close-toed shoes, thus limiting the orthoses use and interfering with the adherence to the treatment. The use of FO will be monitored by phone, but we cannot be sure that they use them every day.

### 3.2. Strengths

This research will have several strengths, such as the random assignment to the treatment and the blinding of the evaluators, the direct applicability of the results obtained and the absence of quality information in this field. This research will clarify many aspects that are still unclear regarding PFF and its treatment. This section may be divided by subheadings. It should provide a concise and precise description of the experimental results, their interpretation, as well as the experimental conclusions that can be drawn.

## Data Availability

No more data is available.

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
