# Peer review of "Efficacy of Personalized Foot Orthoses in Children with Flexible Flat Foot: Protocol for a Randomized Controlled Trial"

_jpm, 2023, doi:10.3390/jpm13081269_

Round 1
Reviewer 1 Report
Congratulations on the study. I consider that it is a very interesting work and that it provides quality information to the scientific community. This protocol can be published after making a few minor changes.
Please justify more clearly the need for this protocol; you should consider including the reference of the study Sagat, P., Bartik, P., Štefan, L. et al. Are flat feet a disadvantage in performing unilateral and bilateral explosive power and dynamic balance tests in boys? A school-based study. BMC Musculoskeletal Disorder 24, 622 (2023). https://doi.org/10.1186/s12891-023-06752-9, for its relationship with your study (This study shows that children with flat feet performed poorer in some physical performance tasks, compared to the normal feet counterparts).
In addition, you should also consider including the reference ZHANG Jin, JIANG Shuyun, LI Yang, YU Yan, LU Xiaoying, LI Yiying. Advantes in diagnosis, prevention and treatment of flexible flatfoot in children[J]. CHINESE JOURNAL OF SCHOOL HEALTH, 2023, 44(6): 946-950., since it is one of the latest studies published on advances in the diagnosis, prevention and treatment of flexible flatfoot in children.
Finally, the part of the protocol where future lines of research are mentioned should be more extensive.
Author Response
Congratulations on the study. I consider that it is a very interesting work and that it provides quality information to the scientific community. This protocol can be published after making a few minor changes.
R. Dear Reviewer,
Thank you for giving us the possibility to address all the comments that came up during the review process. Below you will find all the comments and answers point by point. In the new revised version, changes are highlighted in red.
Please justify more clearly the need for this protocol; you should consider including the reference of the study Sagat, P., Bartik, P., Štefan, L. et al. Are flat feet a disadvantage in performing unilateral and bilateral explosive power and dynamic balance tests in boys? A school-based study. BMC Musculoskeletal Disorder 24, 622 (2023). https://doi.org/10.1186/s12891-023-06752-9, for its relationship with your study (This study shows that children with flat feet performed poorer in some physical performance tasks, compared to the normal feet counterparts).
R. The following information and reference have been added: Sagat, P et al. have shown that children with flat feet presented poor performance in certain physical tasks, in contrast to the control group, with neutral feet [31].
31. Sagat, P., Bartik, P., Štefan, L. et al. Are flat feet a disadvantage in performing unilateral and bilateral explosive power and dynamic balance tests in boys? A school-based study. BMC Musculoskeletal Disorder. 2023, 24, 622.
In addition, you should also consider including the reference ZHANG Jin, JIANG Shuyun, LI Yang, YU Yan, LU Xiaoying, LI Yiying. Advantes in diagnosis, prevention and treatment of flexible flatfoot in children[J]. CHINESE JOURNAL OF SCHOOL HEALTH, 2023, 44(6): 946-950., since it is one of the latest studies published on advances in the diagnosis, prevention and treatment of flexible flatfoot in children.
R. The following sentence and reference have been added: Furthermore, as the authors Zhang J. et al. pointed out, an early identification of the PFF is necessary, thus an intervention plays a crucial role in enhancing the outlook. Also, there is an absence of consistent quantitative standards for diagnosing flexible flatfoot [32].
32. Zhang J., Jiang S., Li Y., Yu Y., Lu X., Li Y. Advantes in diagnosis, prevention and treatment of flexible flatfoot in children[J]. Chinese journal of school health, 2023, 44(6): 946-950.
Finally, the part of the protocol where future lines of research are mentioned should be more extensive.
R. The following information has been added at the end of the “Expected Results” section: The main future research which is required after this protocol study is to carry out the detailed RCT described in the present manuscript. Also, qualitative research to understand the experience of patients with PFF wearing FO is required.
Reviewer 2 Report
I am truly impressed by the depth and significance of the research presented in the article titled "Efficacy of Personalized Foot Orthoses in Children with Flexible Flat Foot: Protocol for a Randomized Controlled Trial." This protocol outlines a comprehensive investigation into a pressing issue in the field of pediatric podiatry, and I find it to be a substantial contribution to the understanding and treatment of flexible flat foot (PFF).
The abstract concisely introduces the complexities surrounding PFF diagnosis and treatment, highlighting the absence of a clear approach and the limited evidence supporting the use of foot orthoses (FOs). The overarching objective to assess the efficacy of personalized FOs alongside specific exercises compared to exercises alone, through a rigorous randomized controlled trial (RCT), is both scientifically sound and practically relevant.
The introduction sets the context and rationale for the study, addressing the prevalence and concerns associated with PFF. It effectively highlights the lack of consensus on treatment and diagnosis, paving the way for the importance of this investigation. The discussion of the biomechanical alterations and compensations occurring in PFF provides a strong foundation for the study's objectives.
The inclusion of specific eligibility criteria ensures that the study focuses on the target population, and the interventions are well-defined in terms of custom insoles and exercise regimens. The detailed explanation of the interventions facilitates replicability, and the focus on blinding participants and evaluators further strengthens the study's scientific rigor.
The article's method section is thorough and meticulously organized, covering aspects such as study design, setting, eligibility criteria, interventions, outcome measures, sample size calculation, and statistical analysis. The transparency in methodology enhances the credibility and replicability of the study.
The strengths and limitations sections offer a balanced perspective on the study. The acknowledgement of potential challenges and limitations, such as patient adherence and climatic conditions affecting orthotic use, reflects the author's thorough consideration of real-world scenarios.
One of the most noteworthy aspects of the study is its potential to provide a standardized definition of PFF based on the common findings presented by the sample. This achievement would be a significant advancement in the field and could lead to improved diagnostic accuracy.
In conclusion, the scientific article excels in its comprehensive approach to addressing a clinically relevant issue, and the detailed protocol for the randomized controlled trial is highly commendable. The potential for this study to contribute valuable evidence for the treatment and diagnosis of PFF, as well as its meticulous methodology, positions it as a substantial and impactful addition to pediatric podiatry research
Author Response
I am truly impressed by the depth and significance of the research presented in the article titled "Efficacy of Personalized Foot Orthoses in Children with Flexible Flat Foot: Protocol for a Randomized Controlled Trial." This protocol outlines a comprehensive investigation into a pressing issue in the field of pediatric podiatry, and I find it to be a substantial contribution to the understanding and treatment of flexible flat foot (PFF).
The abstract concisely introduces the complexities surrounding PFF diagnosis and treatment, highlighting the absence of a clear approach and the limited evidence supporting the use of foot orthoses (FOs). The overarching objective to assess the efficacy of personalized FOs alongside specific exercises compared to exercises alone, through a rigorous randomized controlled trial (RCT), is both scientifically sound and practically relevant.
The introduction sets the context and rationale for the study, addressing the prevalence and concerns associated with PFF. It effectively highlights the lack of consensus on treatment and diagnosis, paving the way for the importance of this investigation. The discussion of the biomechanical alterations and compensations occurring in PFF provides a strong foundation for the study's objectives.
The inclusion of specific eligibility criteria ensures that the study focuses on the target population, and the interventions are well-defined in terms of custom insoles and exercise regimens. The detailed explanation of the interventions facilitates replicability, and the focus on blinding participants and evaluators further strengthens the study's scientific rigor.
The article's method section is thorough and meticulously organized, covering aspects such as study design, setting, eligibility criteria, interventions, outcome measures, sample size calculation, and statistical analysis. The transparency in methodology enhances the credibility and replicability of the study.
The strengths and limitations sections offer a balanced perspective on the study. The acknowledgement of potential challenges and limitations, such as patient adherence and climatic conditions affecting orthotic use, reflects the author's thorough consideration of real-world scenarios.
One of the most noteworthy aspects of the study is its potential to provide a standardized definition of PFF based on the common findings presented by the sample. This achievement would be a significant advancement in the field and could lead to improved diagnostic accuracy.
In conclusion, the scientific article excels in its comprehensive approach to addressing a clinically relevant issue, and the detailed protocol for the randomized controlled trial is highly commendable. The potential for this study to contribute valuable evidence for the treatment and diagnosis of PFF, as well as its meticulous methodology, positions it as a substantial and impactful addition to pediatric podiatry research
- Dear Reviewer,
Thank you for your comments. We really appreciate your feedback and we are looking forward to start the protocol study and share with the research community our findings.